

# Patient-reported outcomes measured with and without dizziness associated with non-specific chronic neck pain: implications for primary care

Raúl Ferrer-Peña[1,2,3], Gonzalo Vicente-de-Frutos[1], Diego Flandez-Santos[1], Carlos Martín-Gómez[1], Carolina Roncero-Jorge[4] and César Calvo-Lobo[5]

[1] Department of Physiotherapy, Facultad de Ciencias de la Salud, Centro Superior de Estudios Universitarios La Salle, Universidad Autónoma de Madrid, Madrid, Spain
[2] Motion in Brains Research Group, Instituto de Neurociencias y Ciencias del Movimiento (INCIMOV), Centro Superior de Estudios Universitarios La Salle, Universidad Autónoma de Madrid, Madrid, Spain
[3] Centro de Salud Entrevías, Gerencia de Atenciín Primaria, Fundación para la Investigación e Innovación Biomédica en Atención Primaria de la Comunidad de Madrid (FIIBAP), Servicio Madrileño de Salud, Madrid, Spain
[4] Servicio de Hospitalización Médica, Hospital Universitario del Sureste, Arganda del Rey, Madrid, Spain
[5] Institute of Biomedicine (IBIOMED), Nursing and Physical Therapy Department, Universidad de León, Ponferrada, León, España

Corresponding authors
Raúl Ferrer-Peña,
raul.ferrer@lasallecampus.es
César Calvo-Lobo,
cecalvo19@hotmail.com

## ABSTRACT

**Background**. The aim of this study was to compare health-related quality of life (HRQoL) and disability and fear of movement in patients with non-specific chronic neck pain (NSCNP) associated with dizziness with respect to patients with isolated NSCNP in primary care settings.

**Methods**. A cross-sectional study was carried out in a primary care center. A total of 120 patients were divided into two groups and analyzed in this study. One group of patients reported dizziness combined with NSCNP ($n = 60$), and the other reported no dizziness with their NSCNP ($n = 60$). Patient-reported outcome measurements were HRQoL (primary outcome) and disability and kinesiophobia (secondary outcomes) assessed by the EuroQoL Five Dimensions and Five Levels (EQ-5D-5L), neck disability index (NDI) and Tampa Scale of Kinesiophobia (TSK-11), respectively.

**Results**. Statistically significant differences ($P < 0.05$) for a 95% confidence interval (CI) with a large effect size (Cohen $d$) were found between both groups with greater values of disability (mean difference = 6.30 points; 95% CI [3.84–8.75]; $d = 0.94$) and kinesiophobia (mean difference = 8.36 points; 95% CI [6.07–10.65]; $d = 1.33$), and an impairment of HRQoL (mean difference = 16.16 points; 95% CI [11.09–21.23]; $d = 1.16$), for patients with NSCNP associated with dizziness with respect to patients with isolated NSCNP.

**Conclusions**. Patients with NSCNP in conjunction with dizziness present higher HRQoL impairment and higher disability and kinesiophobia compared to patients with isolated NSCNP.

## INTRODUCTION

Worldwide, non-specific chronic neck pain (NSCNP) is defined as chronic pain (at least three months in duration) in the neck region without a specific origin with or without referred pain to upper limbs. This non-specific origin may be secondary to the difficulties in establishing an accurate pathological diagnosis, the role of imaging tests cannot explain the symptoms as well as its multidimensional nature (*Huisstede et al., 2007*; *Matsumoto et al., 2010*). This condition is considered a common musculoskeletal condition and has a substantial impact on patient-reported outcome measurements such as health-related quality of life (HRQoL) and functionality (*Hoy et al., 2014*; *GBD 2015 Disease and Injury Incidence and Prevalence Collaborators, 2016*). Indeed, 22% of primary care patients reported persistent pain and are more likely to suffer from activity limitations and unfavorable health perceptions (*Gureje et al., 1998*).

Dizziness is considered a common and challenging condition that is observed in primary care settings and has often been reported as one of the major symptoms associated with NSCNP (*Wipperman, 2014*; *Iglebekk, Tjell & Borenstein, 2017*). In addition, patients with NSCNP may suffer from cervicogenic dizziness with a frequency similar to patients who suffer from cervical spine degeneration and whiplash (*Wrisley et al., 2000*; *Treleaven, Jull & Sterling, 2003*; *Kristjansson & Treleaven, 2009*; *Hain, 2015*). Cervicogenic dizziness is a common condition that seems to lead to complications such as instability and misbalance combined with disability and HRQoL impairment (*Wrisley et al., 2000*; *Reid & Rivett, 2005*; *Weidt et al., 2014*; *Grande-Alonso et al., 2018a*).

Cervicogenic dizziness may be defined as a non-specific anomalous spatial sensation generated by anomalous afferences from the cervical spine. Indeed, it is known that cervicogenic dizziness may be modified secondary to visual, vestibular, and somatosensory system inputs, which control body balance and stability and whose alterations may produce neck pain, headache, and/or muscle stiffness (*Minguez-Zuazo et al., 2016*). Nevertheless, a differential diagnosis should be considered between cervicogenic dizziness (*Devaraja, 2018*) and several clinical entities dealing with specific vestibular alterations such as whiplash and associated disorders (*Treleaven et al., 2016*) or Benign Paroxysmal Positional Vertigo (BVVP) (*Argaet, Bradshaw & Welgampola, 2019*).

Thus, patients who suffer from NSCNP-associated cervicogenic dizziness seem to acquire maladaptive beliefs such as fear of movement, perceived disability, and postural motor control alterations (*Dickin, 2010*; *Alahmari et al., 2014*; *Minguez-Zuazo et al., 2016*; *Grande-Alonso et al., 2018a*). It is known that patients with NSCNP (*Hoy et al., 2014*; *GBD 2015 Disease and Injury Incidence and Prevalence Collaborators, 2016*) and patients with dizziness (*Wrisley et al., 2000*; *Weidt et al., 2014*) present an impaired HRQoL and are some of the most frequently seen patients among primary care patients (*Gureje et al., 1998*; *Wipperman, 2014*). Nevertheless, the effects of dizziness added to NSCNP have not yet been addressed in primary care patients regarding patient-reported outcome measurements such as HRQoL, disability, and kinesiophobia. Indeed, dizziness may impair both physical and mental HRQoL domains due to its association with clinical symptoms and psychosocial factors remaining as a challenge to orthopaedic and vestibular rehabilitation specialists

(*Wrisley et al., 2000*; *Weidt et al., 2014*). We hypothesized that patients with NSCNP in conjunction with dizziness would present higher HRQoL impairment in addition to higher levels of disability and kinesiophobia. Therefore, the main aim of this study was to compare HRQoL as the primary outcome and disability and fear of movement as secondary outcomes in patients with NSCNP associated with dizziness with respect to patients with isolated NSCNP in primary care settings. In addition, the secondary objective was to correlate HRQoL, disability and fear of movement outcome measurements for both patients with NSCNP associated with dizziness and patients with isolated NSCNP.

## METHODS

### Study design

A descriptive, cross-sectional design using non-probability convenience sampling was done. All participants were recruited from September 2017 until February 2018. Patients must have been diagnosed as NSCNP by their referring general practitioner from two primary healthcare centers in Madrid (Spain) and also had been referred to the Physical Therapy Unit. All subjects gave their written informed consent. This study was approved by the Southeast Local Research Committee of the Primary Health Care Management (Code 07/17). The study reporting followed the ''Strengthening the Reporting of Observational studies in Epidemiology'' (STROBE) guidelines (*Von Elm et al., 2007*).

### Subjects

The study included two groups of patients with NSCNP: (1) those who perceived dizziness combined with NSCNP and (2) those who only had NSCNP. All of them were recruited with frequency matching pairing methods (*Setia, 2016*). The selection of the sample was carried out by a physical therapist with more than 13 years of clinical experience based on the clinical record and diagnoses provided by a primary care physician. Inclusion criteria for study participants consisted of several parameters: (1) referred to the Primary Health Physical Therapy Unit after complaining of neck pain; (2) having pain for at least three months or longer; (3) a pain intensity greater than 30 mm on a Visual Analogue Scale (VAS); (4) between 18 and 75 years old; and (5) (specific to study group) record of dizziness associated with their neck pain in the assessment date. The exclusion criteria consisted of several parameters: (1) patients who presented systemic illness (such as rheumatoid arthritis, cancer, and/or fibromyalgia); (2) history of cervical region traumas or surgeries; (3) presence of diagnosed radiculopathies or cervical myelopathies; or (4) presently undergoing treatment or having received any type of treatment during the last three months.

### Data collection

A sociodemographic questionnaire asking about gender, age, height, weight, body mass index (BMI), civil state, education level, chronicity time, pain intensity, and the actual presence of perceived dizziness was administered to all study participants. The patient-reported outcome measurements were the HRQoL (primary outcome) in addition to pain intensity, disability, and kinesiophobia (secondary outcomes).

## Primary outcome
### Health related quality of life
HRQoL was measured using the Spanish version of the EuroQoL Five Dimensions and Five Levels (EQ-5D-5L). This questionnaire has been widely used in the literature to report perceived HRQoL for many health-related conditions (*Herdman et al., 2011*; *Obradovic, Lal & Liedgens, 2013*; *Oppe et al., 2014*). The instrument consists of two elements. The first one is a 5-item questionnaire with one item for each assessed domain (mobility, self-care, usual activities, pain/discomfort, and anxiety/depression) and five levels on each domain (no, slight, moderate, severe, or extreme problems). Patients were asked to fill out only one level for each domain (1 to 5). The digits on each domain were then combined into a 5-digit number ranging from 11,111 to 55,555. In order to interpret the results on the EQ-5D-5L, we used the EuroQoL Index (*EQ-Index*). This approach compares the values in the five dimensions with 3,125 different hypothetic health states adjusted by country population with the "0" value assigned to death and "1" as the perfect health status. Values less than 0 are considered in the index, and those statuses are interpreted as "worse than being dead". The minimum clinically important difference (MCID) for the EQ-5D was estimated to be a mean of 0.18 points on a scale ranging from 0.03 to 0.52 points (*Coretti, Ruggeri & McNamee, 2014*).

Also, the EQ-5D-5L results can be interpreted according to the HRQoL *Sum Score*, which is a *severity index* obtained from the summation of the levels in each of the instrument's dimension, then subtracting five points, and multiplying the result by five. It results in a new scale ranging from 0 to 100 points on which more points indicate more severity. The other side of the instrument consists of a vertical 20 cm *EQ-VAS* on which subjects are asked to self-rate their health, from 0 "The worst health you can imagine" to 100 "The best health you can imagine". Finally, psychometric properties of the EQ-5D-5L showed a minimal floor and ceiling effect (<3%), a Cronbach's α of 0.86 and a strong correlation from 0.688 to 0.782 with pain and function scores (*Bilbao et al., 2018*).

## Secondary outcomes
### Pain intensity
The average pain intensity from the last seven days was measured using a visual analogue scale (VAS). This scale consists of one horizontal 100 mm line on which the patients must indicate their pain intensity. At the left side of the line the text "No Pain" appears, and on the right side the text "Worst possible pain" appears. This instrument was previously shown to be valid and reliable for measuring pain intensity (*Jensen, Karoly & Braver, 1986*; *Jensen et al., 1999*). The VAS psychometric properties showed an intraclass correlation coefficient (ICC) of 0.97 (95% confidence interval from 0.96 to 0.98) (*Bijur, Silver & Gallagher, 2001*).

### Disability
The neck disability index (NDI) may be considered as a validated 10-item tool with each item ranked on a 6-point scale (*Vernon, 2008*). This questionnaire is the most commonly used tool for NSCNP (*Vernon, 2008*; *Kovacs et al., 2008*; *MacDermid et al., 2009*) consisting of a Spanish cross-cultural adapted and validated version with adequate psychometric properties, showing a Cronbach α of 0.89, an ICC of 0.98 and a Pearson correlation

coefficient ($r$) from 0.65 to 0.89 with pain scales and questionnaires (*Andrade Ortega, Delgado Martínez & Almécija Ruiz, 2010*). Its scores may be divided into five categories: (1) ≤ 4 points for no disability; (2) 5–14 points for mild disability; (3) 15–24 points for moderate disability; (4) 25–34 points for severe disability; and (5) ≥ 35 points for complete disability (*Vernon, 2008*; *Kovacs et al., 2008*; *MacDermid et al., 2009*).

### Kinesiophobia

The fear of movement was self-reported by means of the Spanish-validated 11-item version of the Tampa Scale of Kinesiophobia (TSK), which showed good psychometric properties, with an Cronbach $\alpha$ of 0.78 and an ICC of 0.82, for chronic back pain (*Roelofs et al., 2004*; *Gómez-Pérez, López-Martínez & Ruiz-Párraga, 2011*). Each item may be scored on a 4-point Likert-type scale with the scale ranging from "strongly agree" to "strongly disagree". Total scores varied from 11 to 44; higher scores indicated more fear of movement (*Gómez-Pérez, López-Martínez & Ruiz-Párraga, 2011*).

## Sample size

The sample size was estimated using G*Power 3.1.9.2 for Mac OS X (G*Power© from University of Dusseldorf, Germany) in order to determine a sufficient sample size. The sample size calculation was based on the HRQoL *Sum Score (severity index)* of the EQ-5D-5L as the main outcome measurement (*Herdman et al., 2011*; *Obradovic, Lal & Liedgens, 2013*; *Oppe et al., 2014*), A Student's $t$-test with two groups based on a medium effect size ($d = 0.52$) obtained from a pilot study with 10 subjects, five patients reported dizziness combined with NSCNP ($34.00 \pm 18.84$ points) and five patients reported no dizziness with their NSCNP ($26.00 \pm 15.17$ points), was used to obtain the sample size calculation with a statistical power of 80% using an $\alpha$ error of 0.05 and two tails (*Faul et al., 2007*). Based on the aforementioned assumptions, we estimated a sample size of at least 120 subjects, 60 patients for each group.

## Statistical analysis

All data analyses were performed on SPSS for Mac OS X, Version 22.0 (SPSS Inc., Chicago IL) with an 95% confidence interval (CI). Statistically significant differences were considered $P$-values < 0.05. Descriptive statistics were generated for the sociodemographic data (such as age, BMI, chronicity time, pain intensity, and others). The Kolmogorov-Smirnoff test was performed and showed that the majority of the variables were normally distributed in the sample, so parametric tests were used because of the central limit theorem for samples >50 subjects per group (*Mouri, 2013*; *Hazut et al., 2015*). Continuous variables are presented as mean and standard deviation, and the categorical variables are presented as absolute numbers and relative frequency (percentages).

A chi-squared test was used for a comparison of the other qualitative sociodemographic variables and also for the five dimensions of the EQ-5D-5L comparison between groups. A Student's $t$-test for independent samples was used to compare quantitative sociodemographic variables, the HRQoL *Sum Score*, EQ-VAS variables, neck pain disability, and kinesiophobia between groups. Effect sizes were calculated by Cohen's $d$ and divided into small effect size ($d = 0.20 - 0.49$), medium effect size ($d = 0.50 - 0.79$) and large effect

**Table 1** Description and comparison of quantitative sociodemographic variables for the Dizziness and Non-Dizziness NSCNP groups.

| | Dizziness ($n = 60$) Mean $\pm$ SD | Non Dizziness ($n = 60$) Mean $\pm$ SD | Mean Dif. | IC 95% de la Dif. | | | Effect Size (Cohen's $d$) |
|---|---|---|---|---|---|---|---|
| | | | | Inferior | – | Superior | |
| Age (years) | $51.27 \pm 11.75$ | $47.53 \pm 13.36$ | 3.73 | −0.82 | – | 8.28 | $d = 0.30$ |
| BMI | $25.66 \pm 3.86$ | $24.77 \pm 4.07$ | 0.88 | −0.55 | – | 2.32 | $d = 0.23$ |
| Chronicity Time (months) | $43.51 \pm 59.56$ | $55.02 \pm 67.07$ | −11.50 | −34.44 | – | 11.42 | $d = -0.18$ |
| Pain Intensity (VAS) | $63.00 \pm 15.85$ | $58.35 \pm 19.67$ | 4.65 | −1.81 | – | 11.11 | $d = 0.26$ |

**Notes.**

SD, Standard Deviation; CI, Confident Interval; BMI, Body Mass Index; VAS, Visual Analogue Scale.

size ($d > 0.80$) (*Kelley & Preacher, 2012*). Regarding both groups, univariate correlation analyses were performed by the Pearson's correlation coefficient and categorized as weak ($r = 0.00 - 0.40$), moderate ($r = 0.41 - 0.69$) or strong ($r = 0.70 - 1.00$) correlations (*Sedgwick, 2012*).

# RESULTS

A total of 120 patients were divided into two groups and analyzed in this study. One group of patients reported dizziness combined with NSCNP ($n = 60$), and the other reported no dizziness with their NSCNP ($n = 60$). Sociodemographic and descriptive data did not show any statistically significant differences ($P > 0.05$) between groups. Descriptive variables of both groups are presented in Tables 1 and 2.

## Primary outcomes

The EQ-index analysis revealed a statistically significant difference between groups ($P < 0.001$) based on the Student's $t$-test comparison with $0.77 \pm 0.16$ and $0.54 \pm 0.21$ (mean $\pm$ SD) for the non-dizziness and dizziness groups, respectively.

The chi-squared test compared the individual domains and levels of the EQ-5D-5L between groups and showed statistical differences ($P < 0.05$) in the five domains presenting higher levels for the dizziness group with respect to the non-dizziness group (Table 3).

## Secondary outcomes

Regarding the Student's $t$-test, a statistical difference ($P < 0.05$) was found between both groups with respect to the quantitative variables, showing greater neck pain disability and kinesiophobia values and also lower values of HRQoL variables in the dizziness group versus the non-dizziness group. The results of the comparison for the other quantitative variables are presented in Table 4.

## Correlation analyses

The Pearson's correlation analysis revealed statistically significant positive correlations in the dizziness group for the HRQoL Sum Score with the perceived disability ($P < 0.001$; $r = 0.649$), and moderate positive correlations with pain intensity and kinesiophobia. Also, the analyses revealed statistically significant correlations between all analyzed variables. The results are presented in Table 5.

**Table 2  Description of qualitative sociodemographic variables between Dizziness and Non-Dizziness NSCNP groups.**

| | | Dizziness ($n = 60$) n (%) | Non-Dizziness ($n = 60$) n (%) | *P*-Value (Chi-square test) |
|---|---|---|---|---|
| Gender | | | | |
| | Male | 16 (26.7%) | 21 (35%) | 0.429 |
| | Female | 44 (73.3%) | 39 (65%) | |
| Civil State | | | | |
| | Single | 1 (1.7%) | 1 (1.7%) | |
| | Married | 40 (66.7%) | 41 (68.3%) | 0.908 |
| | Divorced | 14 (23.3%) | 15 (25%) | |
| | Widower | 5 (8.3%) | 3 (5%) | |
| Work status | | | | |
| | Unemployed | 7 (11.7%) | 11 (18.3%) | |
| | Active | 39 (65%) | 36 (60%) | |
| | Housewife | 10 (16.7%) | 8 (13.3%) | 0.428 |
| | Retired | 4 (6.7%) | 2 (3.3%) | |
| | Sick Leave | 0 (0%) | 1 (1.7%) | |
| | Student | 0 (0%) | 2 (3.3%) | |
| Education Level | | | | |
| | None | 4 (6.7%) | 3 (5%) | |
| | Primary | 19 (31.7%) | 16 (26.7%) | |
| | Secondary | 14 (23.3%) | 8 (13.3%) | 0.424 |
| | University | 20 (33.3%) | 28 (46.7%) | |
| | Doctorate | 3 (5%) | 5 (8.3%) | |

**Notes.**
All data are presented as number and percentage [n(%)]. * $P < 0.05$

Regarding the non-dizziness group, the Pearson's correlation analyses showed statistically significant positive correlations between the HRQoL Sum Score and the perceived disability ($P < 0.001$; $r = 0.725$) and negative correlations with the self-perceived health status ($P < 0.001$; $r = -0.558$). The results are presented in Table 6.

# DISCUSSION

To the authors' knowledge, this is the first study that has addressed the effects of dizziness and NSCNP together in primary care patients who presented an impairment in patient-reported outcome measurements such as the HRQoL, disability levels, and kinesiophobia. According to the mentioned neurophysiological model (*Minguez-Zuazo et al., 2016*; *Devaraja, 2018*), the hypothesis of this model resides in the fact that the alteration of a sensory input, specially provided by the neurophysiological pathways between the upper cervical region and the subsystems that form the postural control, could cause sensations of dizziness secondary to an integration incongruity between the aberrant cervical somatosensory input and the expected physiological sensory patterns (*Kristjansson & Treleaven, 2009*). Indeed, vestibular and somatosensory systems could compensate for alterations in balance by increasing the rigidity of the body, mainly in the cervical region,

**Table 3** Descriptive data and of five dimensions of EQ-5D and comparison between the Dizziness and Non-Dizziness patients with NSCNP.

| EQ-5D dimension & levels | | Groups | | P-value (Chi-square test) |
|---|---|---|---|---|
| | | Dizziness (n = 60) | Non-Dizziness (n = 60) | |
| Mobility | No problem | 28 (46.7%) | 47 (78.3%) | |
| | Slight problem | 19 (31.7%) | 8 (13.3%) | |
| | Moderate problem | 10 (16.7%) | 5 (8.3%) | 0.003 |
| | Severe problem | 3 (5%) | 0 (0%) | |
| | Unable to | 0 (0%) | 0 (0%) | |
| Self-care | No problem | 30 (50%) | 49 (81.7%) | |
| | Slight problem | 18 (30%) | 6 (10%) | |
| | Moderate problem | 11 (18.3%) | 5 (8.3%) | 0.003 |
| | Severe problem | 1 (1.7%) | 0 (0%) | |
| | Unable to | 0 (0%) | 0 (0%) | |
| Usual activities | No problem | 6 (10%) | 30 (50%) | |
| | Slight problem | 24 (40%) | 17 (28.3%) | |
| | Moderate problem | 25 (41.7%) | 13 (21.7%) | <0.001 |
| | Severe problem | 4 (6.7%) | 0 (0%) | |
| | Unable to | 1 (1.7%) | 0 (0%) | |
| Pain/discomfort | No problem | 0 (0%) | 6 (10%) | |
| | Slight problem | 9 (15%) | 22 (36.7%) | |
| | Moderate problem | 30 (50%) | 26 (43.3%) | <0.001 |
| | Severe problem | 19 (31.7%) | 6 (10%) | |
| | Unable to | 2 (3.3%) | 0 (0%) | |
| Anxiety/depression | No problem | 14 (23.3%) | 32 (53.3%) | |
| | Slight problem | 17 (28.3%) | 15 (25%) | |
| | Moderate problem | 14 (23.3%) | 9 (15%) | 0.004 |
| | Severe problem | 12 (20%) | 4 (6.7%) | |
| | Unable to | 3 (5%) | 0 (0%) | |

**Notes.**
All data are presented as number and percentage [n(%)].
NSCNP, Non-Specific Chronic Neck Pain; EQ-5D, EuroQoL Five Dimensions.

**Table 4** Comparison of quantitative variables for the Dizziness and Non-Dizziness groups in patients with NSCNP.

| | Dizziness (n = 60) Mean ± SD | Non-Dizziness (n = 60) Mean ± SD | Mean Dif. | 95% of CI | | | Effect Size (Cohen's d) |
|---|---|---|---|---|---|---|---|
| | | | | Inferior | – | Superior | |
| HRQoL Sum Score | 34 ± 14.34 | 17.83 ± 13.69 | 16.16 | 11.09 | – | 21.23 | $d = 1.16^*$ |
| EQ VAS | 55.25 ± 17.76 | 67.97 ± 19.77 | −12.71 | −19.51 | – | −5.92 | $d = -0.68^*$ |
| NDI | 19.08 ± 7.55 | 12.78 ± 5.91 | 6.30 | 3.84 | – | 8.75 | $d = 0.94^*$ |
| TSK-11 | 32.40 ± 6.56 | 24.03 ± 6.10 | 8.36 | 6.07 | – | 10.65 | $d = 1.33^*$ |

**Notes.**
NSCNP, Non-Specific Chronic Neck Pain; SD, Standard Deviation; CI, Confident Interval; VAS, Visual Analogue Scale; HRQoL, Health Related Quality of Life; EQ VAS, EuroQoL Visual Analogue Scale; NDI, Neck Disability Index; TSK-11, Tampa Scale of Kinesiophobia.
*$P < 0.05$

**Table 5  Pearson's correlations with continuous variables in the dizziness group.**

|  | Pain intensity | NDI | TSK | VAS-EQ | HRQoL SumScore |
|---|---|---|---|---|---|
| Pain intensity | 1 | 0.352[**] | 0.278[*] | −0.558[**] | 0.451[**] |
| NDI | 0.352[**] | 1 | 0.526[**] | −0.558[**] | 0.649[**] |
| TSK | 0.278[*] | 0.526[**] | 1 | −0.479[**] | 0.471[**] |
| VAS EuroQoL | −0.558[**] | −0.558[**] | −0.479[**] | 1 | −0.679[**] |
| HRQoL SumScore | 0.451[**] | 0.649[**] | 0.471[**] | −0.679[**] | 1 |

Notes.

HRQoL, Health Related Quality of Life; NDI, Neck Disability Index; TSK, Tampa Scale of Kinesiophobia; VAS-EQ, Visual Analogue Scale from EuroQoL-5D-5L.

[*]The correlation is statistically significant at an alpha level of 0.05 (2 tails).

[**]The correlation is statistically significant at an alpha level of 0.01 (2 tails).

**Table 6  Pearson's Correlations with continuous variables in non-dizziness group.**

|  | Pain Intensity | NDI | TSK | VAS EuroQoL | HRQoL SumScore |
|---|---|---|---|---|---|
| Pain Intensity | 1 | 0.340[**] | 0.112 | −0.225 | 0.325[*] |
| NDI | 0.340[**] | 1 | 0.147 | −0.527[**] | 0.725[**] |
| TSK | 0.112 | 0.147 | 1 | −0.301[*] | 0.249 |
| VAS-EQ | −0.225 | −0.527[**] | −0.301[*] | 1 | −0.558[**] |
| HRQoL SumScore | 0.325[*] | 0.725[**] | 0.249 | −0.558[**] | 1 |

Notes.

HRQoL, Health Related Quality of Life; NDI, Neck Disability Index; TSK, Tampa Scale of Kinesiophobia; VAS-EQ, Visual Analogue Scale from EuroQoL-5D-5L.

[*]The correlation is statistically significant at an alpha level of 0.05 (2 tails).

[**]The correlation is statistically significant at an alpha level of 0.01 (2 tails).

thus explaining the hyperactivity of the cervical musculature in patients with cervicogenic dizziness (*Minguez-Zuazo et al., 2016*; *Devaraja, 2018*). Nevertheless, patients with NSCNP presented an alteration in cervical neuro-sensorimotor control and, conversely, these patients did not present dizziness (*Falla, Bilenkij & Jull, 2004*). This issue may be explained by the presence of dizziness could be mediated by an amplified central sensitization process (*Holle et al., 2015*).

Recently, the cervico-ocular reflex was shown to be increased in both patients with chronic non-traumatic neck pain without dizziness and patients with chronic traumatic neck pain due to whiplash with associated dizziness. The role of reflexes should be compared in patients who suffer from NSCNP with or without associated dizziness for future studies (*Ischebeck et al., 2018*). In addition, a very relevant aspect in cervical dizziness which was not measured in the present study is the cervical movement. Some authors consider a hypothesis that goes towards the term "dizziness evoked by cervical movements" would be more appropriate than the current neurophysiological hypothesis of "cervicogenic" dizziness. Cervical dizziness may cause a reduction in cervical mobility in general terms, and specifically, cervical rotation movement (*Reid et al., 2014a*; *L'Heureux-Lebeau et al., 2014*; *Williams et al., 2017*; *Grande-Alonso et al., 2018a*).

## HRQoL

EQ-index data showed worse outcomes in patients with combined dizziness and neck pain than in patients with only neck pain. On the other hand, after analyzing the EQ-5D-5L with an individual comparison between domains and levels, we also found worse results for patients with combined dizziness and neck pain than for those with only neck pain. Our results support prior studies addressing HRQoL in patients with NSCNP (*Hoy et al., 2014*; *GBD 2015 Disease and Injury Incidence and Prevalence Collaborators, 2016*) in addition to patients with dizziness (*Wrisley et al., 2000*; *Weidt et al., 2014*), but our findings add the negative HRQoL effects due to dizziness in conjunction with NSCNP. In addition, neck pain and dizziness are common conditions that frequently appear in patients who attend to primary healthcare services (*Gureje et al., 1998*; *Wipperman, 2014*). Thus, dizziness is a persistent condition that may be associated the persistent back pain that is often found in primary care environments (*Gureje et al., 1998*; *Tschan et al., 2013*; *Wipperman, 2014*; *Iglebekk, Tjell & Borenstein, 2017*). Consequently, this condition may produce a high economic burden to primary healthcare services (*Sun et al., 2014*; *Mueller et al., 2014*; *Weidt et al., 2014*).

## Disability and fear of movement

The variables of perceived disability in neck pain and kinesiophobia demonstrated worse scores for patients with combined neck pain and dizziness as opposed to those whose only ailment was neck pain. Dizziness may predispose an individual with NSCNP to development of instability and misbalance in addition to an increase in disability levels (*Wrisley et al., 2000*; *Reid & Rivett, 2005*; *Dickin, 2010*; *Alahmari et al., 2014*; *Weidt et al., 2014*; *Minguez-Zuazo et al., 2016*; *Grande-Alonso et al., 2018a*), which may lead to a higher rate of falls, disability-related problems, and economic burden in primary care patients (*Sun et al., 2014*; *Mueller et al., 2014*; *Weidt et al., 2014*). Therefore, patients who suffered from dizziness associated to NSCNP showed greater fear of movement than patients with isolated NSCNP, which seemed to be in line with prior studies (*Dickin, 2010*; *Alahmari et al., 2014*; *Minguez-Zuazo et al., 2016*; *Grande-Alonso et al., 2018a*). This fact may be due to dizziness may produce a non-specific anomalous spatial sensation secondary to anomalous spine afferences which may alter visual, vestibular or somatosensory inputs (*Minguez-Zuazo et al., 2016*).

Regarding disability and kinesiophobia of our study, patients with NSCNP without cervicogenic dizziness presented a mean of 55 months of chronicity while patients with dizziness associated to NSCNP showed 43 months suffering from this condition. Despite there were not statistically significant differences between both groups, this fact could suggest greater differences in outcome measurements such as disability and fear of movement. Nevertheless, kinesiophobia scores showed a mean of 24 points in the non-dizziness group while patients with dizziness displayed a mean of 32 points. Regarding chronic pain, the minimum clinically important difference for the TSK-11 was set at 4.8 points and thus the difference of 8 points between both groups may be considered as clinically relevant (*Woby et al., 2005*; *George, Valencia & Beneciuk, 2010*; *Hapidou et al., 2012*). In addition, the NDI showed mild disability in the non-dizziness group while patients

with dizziness presented moderate disability (*Vernon, 2008*; *Kovacs et al., 2008*; *MacDermid et al., 2009*). The values of the non-dizziness group could be low for disability and fear of movement compared to the chronicity time showed by these patients. Nevertheless, the presence of dizziness could influence a greater impairment of disability and kinesiophobia, even with less time of chronicity, due to a possible wider central sensitization process (*Holle et al., 2015*).

## Outcome measurements correlations

Regarding correlation analyses of our study, all outcome measurements showed statistically significant correlations between them, except for the severity of health quality of life (HRQoL SumScore) and Kinesiophobia (TSK) in the non-dizziness group. Other previous studies have addressed the relationship between HRQoL and psychological outcome measurements associated with pain perpetuation, such as perceived disability and kinesiophobia. According to *Kovacs et al. (2004)* the association between these outcome measurements was analyzed in patients with chronic low back pain showing similar correlation findings in line with our study.

As mentioned above, in our study, kinesiophobia was the only outcome measure that did not show any statistically significant correlation with the quality of life severity index in the group of patients without dizziness. This finding could be due to kinesiophobia may be an outcome measure with less influence in subjects with neck pain who did not present associated dizziness. Indeed, kinesiophobia did not present statistically significant correlations with pain intensity or neck disability. Nevertheless, kinesiophobia did seem to be an key outcome measure in those patients with cervicogenic dizziness according to the correlations analysis in the group that presented dizziness and in the same line with previous studies evaluating postural control, psychological and disability variables in patients with cervicogenic dizziness and neck pain (*Grande-Alonso et al., 2018b*).

## Implications for primary care

These results provide a better understanding of the combined effect of these two pathologies, demonstrating the great impact it has on the HRQoL. The area in which this study was conducted was primary care, which is of vital importance since it is the gateway to the public healthcare system in which chronicity and pathology of diseases can best be managed with a correct assessment, referral to specialists, and early intervention (*Gureje et al., 1998*; *Wipperman, 2014*). It is important for nurses and clinicians to evaluate all dimensions of the pathology since only by using a biopsychosocial approach can we understand the patient's condition and intervene satisfactorily, thus avoiding the progression of high chronicity with its associated socio-sanitary costs given the high perceived disability that we report in this work (*Valenzuela-Pascual et al., 2015*).

## Future studies

Due to the reported findings, new interventions should be administered to patients who have combined dizziness and NSCNP in primary care settings in order to increase patients' HRQoL, functionality, and stability. Some treatments such as soft tissue manual therapy (*Reid & Rivett, 2005*; *Reid et al., 2014b*), joint mobilization (*Reid et al., 2014b*), therapeutic

patient education, and exercise therapy (*Minguez-Zuazo et al., 2016*) form a multimodal approach that includes nursing, physical therapy, and other medical specialities. These treatments should be compared by means of randomized clinical trials in NSCNP patients with dizziness in primary healthcare services.

### Key practice points

Greater impairment of HRQoL, functionality, and movement sense were shown in patients with NSCNP combined with dizziness compared to patients with isolated NSCNP. Healthcare actions in research, policy, management and/or education regarding primary care environments for minimizing/preventing HRQoL-associated impairment, disability, and kinesiophobia in patients with NSCNP should be taken, especially when it the NSCNP is combined with dizziness. Physicians and physical therapists can provide these questionnaires to patients with NSCNP and dizziness to detect HRQoL impairment, disability, and kinesiophobia in primary care settings and refer these patients to healthcare management specialists. NSCNP and dizziness management/education in primary care settings should be systematically included in multimodal treatment approaches for patients who suffer from both syndromes.

### Limitations

With reference to the study limitations, there may be an inconsistent practice of data transcription. There may also be a limitation with respect to the sample population as patients were recruited from a primary care setting, and although in principle, the results were extrapolated to the entire population given the sample calculation, the conclusion may not be appropriate for certain populations. We also have a high chronicity in both groups within the sample although homogeneity between them is fulfilled. Associated with chronicity is the level of pain severity since one of the inclusion criteria was pain >3 on a VAS scale, which is considered severe/moderate pain. This result was not extrapolated to include patients with mild or no pain. In addition, motion sickness disability was not measured and may be recognized as a main limitation of the present study due to there was not specific outcome measurements related to dizziness. Finally, the present study only evaluated outcome measurements by self-registrations and this may be a limitation due to physical variables, such as cervical range of motion, would be useful in this type of patients in order to study their influence on quality of life and dizziness. Indeed, the TSK-11 total score was the only fear of movement outcome measurement due to this value may be considered as the most relevant index, nevertheless the comparison of the TSK-11 subscales such as Harm and Activity avoidance domains between both groups should be considered for future studies (*Roelofs et al., 2004*; *Gómez-Pérez, López-Martínez & Ruiz-Párraga, 2011*).

### CONCLUSION

Patients with NSCNP combined with dizziness present higher HRQoL impairment in addition to higher disability and kinesiophobia compared to patients who only suffer from isolated NSCNP.

## ACKNOWLEDGEMENTS

The authors did not receive any financial assistance or have any personal relationships with other people or organizations that could inappropriately influence (bias) their work.

### Funding
The authors received no funding for this work.

### Competing Interests
The authors declare there are no competing interests.

### Author Contributions
- Raúl Ferrer-Peña conceived and designed the experiments, performed the experiments, analyzed the data, contributed reagents/materials/analysis tools, prepared figures and/or tables, authored or reviewed drafts of the paper, approved the final draft.
- Gonzalo Vicente-de-Frutos, Diego Flandez-Santos, Carlos Martín-Gómez and Carolina Roncero-Jorge conceived and designed the experiments, performed the experiments, contributed reagents/materials/analysis tools, authored or reviewed drafts of the paper, approved the final draft.
- César Calvo-Lobo conceived and designed the experiments, analyzed the data, prepared figures and/or tables, authored or reviewed drafts of the paper, approved the final draft.

### Human Ethics
The following information was supplied relating to ethical approvals (i.e., approving body and any reference numbers):

This study was approved by the Southeast Local Research Committee of the Primary Health Care Management (Code 07/17).

### Data Availability
The raw measurements are available as a Supplemental File.

### Supplemental Information
Supplemental information for this article can be found online at http://dx.doi.org/10.7717/peerj.7449#supplemental-information.

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
