# Peer review of "Patient-reported outcomes measured with and without dizziness associated with non-specific chronic neck pain: implications for primary care"

_PeerJ, doi:10.7717/peerj.7449_

## Round 0.1 · original submission · Major Revisions

The two reviewers and I feel there is potential in this manuscript, and request the authors attend to these comments.

Reviewer 1 ·

Basic reporting

no comment

Experimental design

no comment

Validity of the findings

no comment

Additional comments

Thank you very much for allowing me to review this manuscript. This work is well designed, well written and offers novel information about a symptom such as dizziness that is very prevalent in patients with non-specific neck pain. I believe that clinicians dedicated to treating patients with neck pain may find the information provided in this study useful.
I believe that the work has great potential for publication in this journal, however, there are several unknowns that must be resolved before publication.
Below are some questions that should be addressed before this work is published:

1. In the summary, add more specific data such as effect sizes, 95% confidence intervals or means of the most relevant results.
2. The introduction is correct, but the hypothesis needs to be better argued.
3. In the material and methods mention who made the selection of the sample based on the inclusion and exclusion criteria.
4. Why was no measure used to measure motion sickness disability? This should be recognized as a limitation of work as there is no specific variable related to dizziness.
5. It is important that psychometric reliability data be entered for each of the questionnaires used in this study.
6. The sample size calculation data that were used do not coincide with the final result of the sample number used, if it is an error correct it.
7. However, the statistical analysis does not explain the procedure of the type of statistical analysis that was performed to calculate the size of the effect. Please include this in the statistical analysis.
8. I believe that in this study a correlation analysis should be performed in order to make the most of the results. Perform the analysis and also discuss the results.
9. In this study results are only obtained from self-registrations, I think this may be a limitation as it would be useful to see the behavior of physical variables in this type of patients and their influence on quality of life and dizziness. Add this to the limitations.
10. The result on fear should be discussed in more depth, review that section.

Reviewer 2 ·

Basic reporting

The present study aimed to compare quality of life and disability among patients with chronic non-specific neck pain with or without associated cervicogenic dizziness. This study has an appropriate design but there are some aspects that should be addressed with the aim of improving the quality of the study. In spite of this, I believe that this is a useful study that provides interesting information in this field.

Experimental design

Sample Size (Line 138)
Is this calculation correct?
I think the authors should check it out.

Validity of the findings

No comment

Additional comments

This study was well described.
-General comments
Introduction section
Line 21-27: NSCNP definition should be reformulated. A brief description of why chronic neck pain is considered mostly nonspecific should be included (e.g: difficulty in establishing an accurate pathological diagnosis, the role of imaging test, multidimensional nature, etc.).
Line 37-41: A brief differentiation between cervicogenic dizziness and several clinical entities dealing with vestibular alterations should be included here (e.g. whiplash and associated disorders, BVVP vertigo (neurophysiological framework)).
Line 52-55: Objectives should be rewritten by please deleting parentheses.
Results section
My main concern in this section is the following:
Patients with NSCNP without cervicogenic dizziness had a mean of 55 months of chronicity (approx.). However, kinesiophobia scores showed a mean of 24 points. In addition, the NDI showed mild disability. I feel that these values are low compared to the chronicity time shown by these patients. This should be discussed.
In addition, Could the authors add the kinesiophobia score that the scientific literature reports as clinically relevant?
Finally, the comparison of the subscales (Harm and Activity avoidance subscales) between the two groups should also include (results section and also in Table 4).
Discussion section
The discussion should be deeply broadened.
Previously, the neurophysiological model has been mentioned. The hypothesis of this neurophysiological model resides in the fact that the alteration of a sensory input, specially provided by the neurophysiological pathways between the upper cervical region and the subsystems that form the postural control, could cause sensations of dizziness. It is therefore that this hypothesis resides in an integration incongruity between the aberrant cervical somatosensory input and the expected physiological sensory patterns. This was reported by Kristjansson and Treleaven a few years ago. Here, vestibular and somatosensory systems could compensate for alterations in balance by increasing the rigidity of the body, mainly in the cervical region, thus explaining the hyperactivity of the cervical musculature in patients with cervicogenic dizziness. However, Deborah Falla, Gwendolen Jull and colleagues (2004) found that patients with NSCNP presented an alteration in cervical neuro-sensorimotor control and conversely, these patients did not present dizziness. All of this should be discussed.
Apart from that, Ischebeck and colleagues (2017) recently have found that the cervico-ocular reflex is increased both, in patients with chronic non-traumatic neck pain without dizziness, and in patients with chronic traumatic neck pain due to whiplash with associated dizziness. The role of reflexes in comparing NSCNP patients with or without dizziness should also be addressed briefly.
In addition to this, a very relevant aspect (extremely important I would dare to say) in cervical dizziness is the cervical movement. I think that in the future, a hypothesis that goes towards the term "dizziness evoked by cervical movements" would be more appropriate than the current neurophysiological hypothesis “cervicogenic”. It has been reported that cervical dizziness may cause a reduction in cervical mobility in general terms, and specifically, cervical rotation movement.
Why was the active cervical range of motion not evaluated in the present study?
I can understand that the objectives were to assess disability and quality of life, but I think that range of motion can also play a relevant role in this variables, don't you think?
I feel this should be considered as a limitation.

---

## Round 0.2 · accepted · Accept

Thanks for addressing all of the initial concerns raised by the reviewers and I.

Reviewer 1 ·

Basic reporting

the article fulfills everything

Experimental design

the article fulfills everything

Validity of the findings

the article fulfills everything

Additional comments

I congratulate the authors for the great work of adapting to the manuscript according to the requests made previously. The manuscript is ready to be published.

Reviewer 2 ·

Basic reporting

no comment

Experimental design

no comment

Validity of the findings

no comment

Additional comments

First of all, I would like to thank the authors for their efforts to address all the issues requested by both reviewers. I think the study is better now and would recommend acceptance for publication.
I would just like to add, although this is already of little relevance, that the cut (value) from which kinesiophobia is considered to be present or not was not added.
Patients with neck pain without dizziness showed less than 28-29/44 points and therefore, we cannot consider them to have kinesiophobia. It would have been interesting to compare patients with neck pain without dizziness who did have high levels of kinesiophobia. It is not clear to me whether this can be considered a limitation, but I just wanted to let the authors know.